# Mesophotic benthic communities associated with a submerged palaeoshoreline in Western Australia

**Mary Wakeford**[1]*, **Marji Puotinen**[1], **William Nicholas**[2], **Jamie Colquhoun**[1], **Brigit I. Vaughan**[3], **Steve Whalan**[1,4], **Iain Parnum**[5], **Ben Radford**[1], **Mark Case**[1], **Ronen Galaiduk**[1], **Karen J. Miller**[1]

1 Australian Institute of Marine Science, Indian Ocean Marine Research Centre, University of Western Australia, Crawley, Western Australia, Australia, 2 Private Contractor, Queanbeyan, New South Wales, Australia, 3 Department of Biodiversity, Conservation and Attractions, Technology Park, Western Precinct, Kensington, Western Australia, Australia, 4 Faculty of Science and Engineering, Southern Cross University, Lismore, New South Wales, Australia, 5 Centre for Marine Science and Technology, Curtin University, Bentley, Western Australia, Australia

* m.wakeford@aims.gov.au

**Data Availability Statement:** All relevant data are within the paper, or accessible from the Australian Institute of Marine Science Data Centre repository:

## Abstract

Key ecological features (KEFs) are elements of Australia's Commonwealth marine environment considered to be important for biodiversity or ecosystem function, yet many KEFs are poorly researched, which can impede effective decision-making about future development and conservation. This study investigates a KEF positioned over the Last Glacial Maximum (LGM) shoreline on the northwest shelf of Australia (known as the 'Ancient Coastline at ~125m depth contour'; AC125). Seafloor bathymetry, sedimentology and benthic habitats were characterised within five study areas using multibeam sonar, sediment samples and towed video imagery. Direct evidence for the existence of a palaeoshoreline formed during the LGM was not found, however candidate areas to find palaeoshoreline material at or just below the modern seabed were discovered. Approximately 98% of the seabed surveyed was comprised of unconsolidated soft sediment habitat (mud/sand/silt) supporting negligible epibenthic biota. The prevalence of soft sediment suggests that post-glacial sediments have infilled parts of the palaeoshoreline, with cross-shelf, probably tidal currents in the northern section of the study area responsible for some of the sediment mobilisation and southern study areas more influenced by oceanic conditions. Within study areas, total biotic cover ranged from 0.02% to 1.07%. Of the biota encountered, most comprised filter feeder organisms (including gorgonians, sponges, and whip corals) whose distribution was associated with pockets of consolidated hard substrate. Benthic community composition varied with both study area and position in relation to the predicted AC125. In general, consolidated substrate was proportionally higher in water shallower than the AC125 compared to on the AC125 or deeper than the AC125. Spatially continuous maps of predicted benthic habitat classes (pre-determined benthic communities) in each study area were developed to characterise biodiversity. Spatial modelling corroborated depth and large-scale structural complexity of the seafloor as surrogates for predicting likely habitat class. This study provides an important assessment of the AC125 and shows that if a distinct coastline exists in the areas

NWSSRP Theme 2 - Project 2a: Characterising the Ancient Coastline Key Ecological Feature (KEF) website (https://apps.aims.gov.au/metadata/view/6b325d80-a983-452a-97e6-39bb723245e6).

**Funding:** AIMS' North West Shoals to Shore Research Program is proudly supported by Santos (www.santos.com) as part of the company's commitment to better understand Western Australia's marine environment. The funders had no role in the study design, data collection and analysis, decision to publish or the preparation of the manuscript.

**Competing interests:** Santos is the commercial funder of this research yet they had no role in the study design, data collection and analysis, the decision to publish or the preparation of the manuscript. This does not alter our adherence to PLOS ONE policies on sharing data and materials.

we surveyed, it is now largely buried and as such does not provide a unique hard substrate habitat. However, much work remains to fully locate and map the ancient coastline within the vast region of the AC125 and additional surveys in shallow waters adjacent to the AC125 may identify whether some sections lie outside the currently defined KEF.

## Introduction

Marine ecosystems cover about 70 percent of the Earth's surface and include the open ocean, the deep-sea ocean, and coastal marine ecosystems; all of which are ecologically, culturally, and economically valuable [1]. Nearly half of the world's population live near the coast [2] and activities such as overfishing, coastal development and pollution can threaten the health and sustainability of these areas. The vulnerability and value of marine ecosystems is recognised through management that contributes to the protection of representative areas which are spatially distributed [3]. Management tools include ecosystem-based approaches that incorporate a holistic approach to managing the entire ecosystem by considering all the links among the living and nonliving resources to extend protection across biota and habitats, with an overarching aim to protect ecological, cultural, and economical values [4]. Effective spatial management that endeavours to conserve and protect biodiversity is grounded in knowledge of biogeographic patterns and habitat heterogeneity within ecosystems [4, 5].

Australian Commonwealth waters are among the largest marine jurisdictions in the world and support significant biodiversity found around mainland Australia, Tasmania, and offshore islands (including sub-Antarctic islands) and include ecosystems such as coral reefs and mangroves of the tropical north to the kelp forests of the temperate south [6, 7]. Many marine ecosystems in Australia are underexplored due to their remoteness and vastness, particularly those in depths greater than 30 m that lie beyond safe SCUBA depths [8]. Marine ecosystems in depths greater than 30 m are exposed to similar pressures that impact shallow water environments including ocean warming and acidification, commercial activities (e.g., oil and natural gas extraction, fishing, tourism) and increased stratification of oceanic layers [9–12].

The management and conservation of Australian Commonwealth waters relies on a spatial framework of bioregions, which are defined by the Integrated Marine and Coastal Regionalisation of Australia (IMCRA) and form the basis of Australia's representative system of marine protected areas [13]. IMCRA targets optimal representation across habitats in marine ecosystems [14, 15], depths, key ecological features (KEFs), biologically informed seascapes, and physical seafloor features [13]. Identifying and protecting representative areas has been challenging in many Australian marine bioregions [16] because of the lack of baseline data which is needed to inform the range of habitat types necessary for optimal ecosystem-level management [16]. Meeting this shortfall relies on data that identifies habitat heterogeneity so that representative samples of the full range of habitats can be protected to optimise biodiversity protection [16, 17].

The North-west Marine Region (NWMR, Fig 1) includes over 1 million km$^2$ of marine habitat, extending from shallow coastal waters to abyssal plain (>5000 m) at the border of Australia's exclusive economic zone [18]. There are 13 KEFs, including canyons down to 2000–3000 m, found within the NWMR which contribute heavily to the conservation values associated with the region [18]. These KEFs are typically characterised by high species richness, high biological productivity, endemism, or unique seafloor geomorphology [18]. Past sea level change has resulted in drowned coastlines around Australia's margin, one of which, at ~125m water

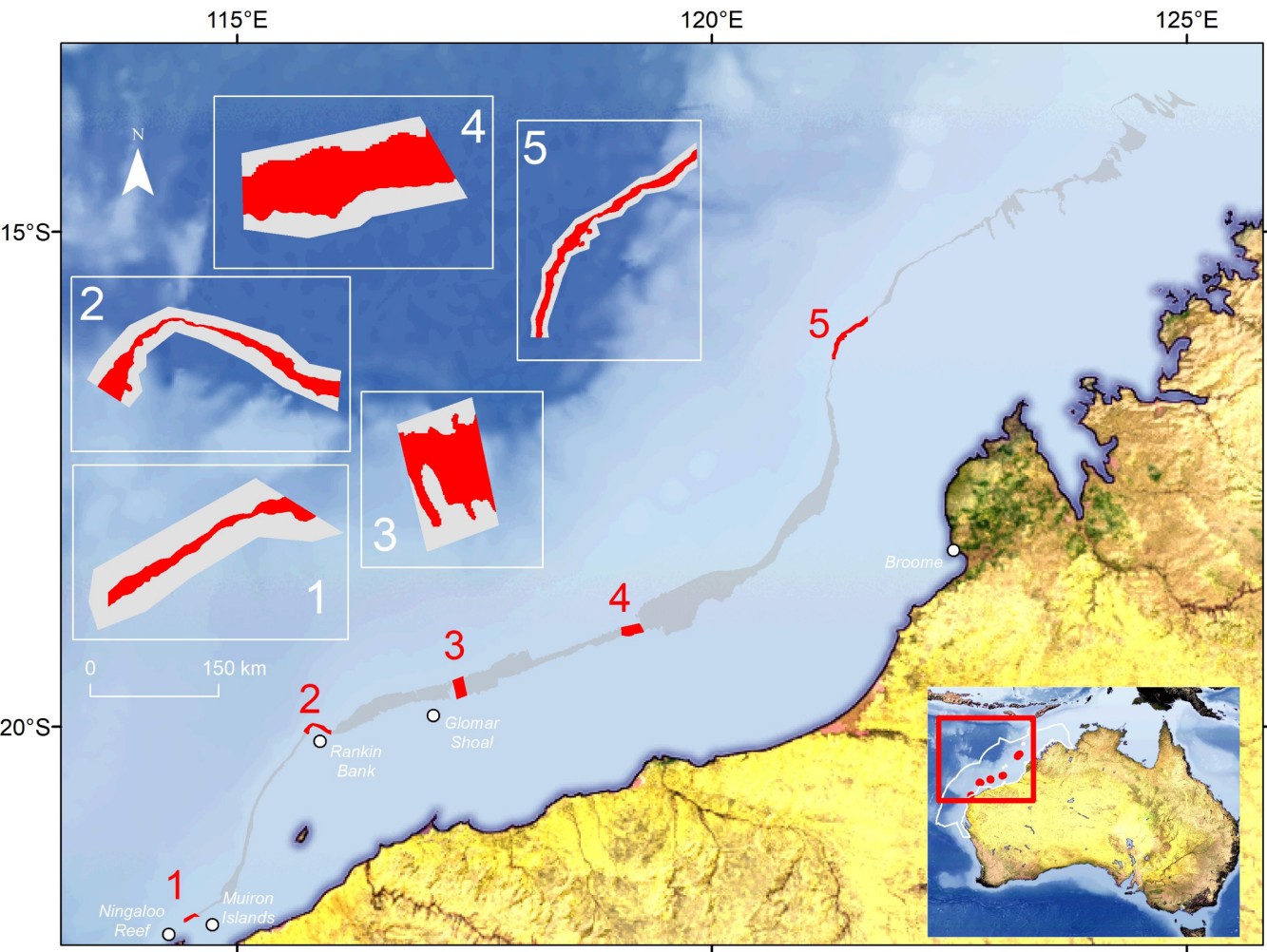

**Fig 1. The ancient coastline KEF at the 125 m depth contour (AC125).** The feature is offshore from northwest Australia (inset map, bottom right). The spatial extent of the AC125 is defined in grey on the main map, with the five study areas highlighted and numbered in red. The five inset maps (top left) show each of the study areas (shaded grey) which include seafloor shallower and deeper than the AC125 (shaded red). The boundary of the North-west Marine Region is shown in white on the inset map (bottom right). NASA Earth Observatory map by Joshua Stevens using data from NASA's MODIS Land Cover, the Shuttle Radar Topography Mission (SRTM), the General Bathymetric Chart of the Oceans (GEBCO), and Natural Earth boundaries.

depth (AC125), is predicted to have formed during the Last Glacial Maximum (LGM), on the northwest shelf [19, 20]. The precise details of the geomorphology and sedimentary composition of modern seabed across the location of the LGM, which the AC125 represents, has been uncertain. The seabed across the LGM shoreline is extensive and is included as a KEF because of the expectation that its seafloor habitats contribute to the ecology of both benthic and pelagic habitats in this region [18]. It is one of the largest and least studied KEFs in the NWMR, spanning over 1500 km and its topographic variation is thought to play a role in altering local oceanographic processes, resulting in nutrient upwelling and associated increased regional productivity [18, 21].

The AC125 can be separated into a northern section (north of Broome; Fig 1) characterised by low primary production and a southern section characterised by high primary production with moderate habitat diversity [22]. Both sections are expected to support communities

representative of the lower-mesophotic zone of the northwest shelf, comprising a mix of hard- and soft-sediment communities. A detailed understanding of the ecological composition of the southern section of the AC125 and how it functions as a KEF, is not well understood [22]. Preserved palaeoshorelines can include seabed features that support high benthic biodiversity [23, 24], particularly when contiguous with contrasting soft sediment habitats that support different faunal communities [25]. Topographical complexity that can be associated with consolidated substrate may provide the critical basis for biodiverse communities of sessile epifauna along the AC125, as in other northern Australian mesophotic ecosystems [26–28]. The AC125 is presumed to contribute to mesophotic biodiversity in the NWMR because it is expected to represent a potential band of seabed topographical structure surrounded by low relief, soft sediment seabed [18, 20, 21]. However, sessile benthic diversity and species richness relative to the surrounding areas of predominantly soft sediment in the AC125 is largely undocumented [18]. This prevents the identification of representative habitat that ideally would underpin management choices. More robust assessment requires detailed information on the characteristics and value of the AC125 in relation to the surrounding areas by determining whether the AC125 represents a distinct seabed type that is important in structuring benthic community composition when compared to areas shallower and deeper than the AC125. To begin to address these knowledge gaps, we aimed to 1) map seafloor bathymetry of the KEF and its surroundings in high resolution using multibeam sonar; 2) determine sediment grainsize and broad composition using benthic grabs; 3) characterise the spatial distribution of benthic communities using towed video, and 4) predict the fine-scale spatial distribution of key habitats across study areas beyond where surveys were undertaken.

## Materials and methods

### Study location

The AC125 extends from North West Cape to the Bonaparte Gulf, Western Australia (Fig 1) and spans the ~115 m to ~135 m depth contours. It repeatedly narrows and widens along its length (minimum width ~ 0.3 km up to 45 km at is widest in the central section). Five study areas were designated along a 1100 km stretch of the AC125. They were chosen to represent a range of expected topographic complexity from relatively flat with gradual depth change (Areas 3 and 4) to relatively steep with abrupt depth change (Areas 1, 2 and 5), and to explore potential latitudinal gradients. Area 1 lies near (~18 km) Ningaloo Reef and the Muiron Islands, both of which support biodiverse mesophotic and coral reef habitats [29, 30]. Biodiverse Rankin Bank [28] lies ~1.5 km from Area 2 and Glomar Shoal (another KEF) is ~30 km from Area 3. Area 5 is north of 17˚S, representing an area of more diverse seascapes [22] and is in a different IMCRA bioregion to the other four areas [31]. Our surveys along the length of the designated AC125 included research in the Kimberly Marine Park which was conducted under an Australian Marine Park Activity Permit (PA2019-00019-1) authorised by the Australian Government Director of National Parks.

Seafloor bathymetry (multibeam sonar), sediment composition (Smith–McIntyre grabs) and benthic community data (towed video) were collected in the five study areas during four voyages between March and October 2019 on the Australian Institute of Marine Science (AIMS) research vessel RV *Solander*. Sampling in each area incorporated surveys on the AC125 (~115–135 m depth), shallower than the AC125 (< 115 m depth) and deeper than the AC125 (> 135 m depth) to determine if the AC125 represented a distinct seabed type and whether position in relation to the AC125 was important in structuring benthic community composition.

## Physical characteristics of the AC125

An R2Sonic 2026 and/or 2022 multibeam echo-sounder were used on the RV *Solander* to survey the seafloor bathymetry in each area. Multibeam survey of Area 1 was undertaken in 2018 during a collaborative research project between Commonwealth Scientific and Industrial Research Organization (CSIRO) and AIMS through the Indian Ocean Marine Research Centre. Multibeam surveys in Areas 2, 3 and 4 were undertaken in 2018–2019 by AIMS as part of the North West Shoals to Shore Research Program (NWSS). Relatively high-resolution bathymetry data already existed for Area 5 as part of the Geoscience Australia 30 m data product for the northwest shelf [32]. For newly acquired multibeam data, depth values between multibeam swaths were estimated using universal block kriging interpolation [33] and the interpolated rasters were gridded at a resolution of a 10 m pixel. Universal block kriging was used rather than a spline or IDW because it models the spatial relationship in the data implicitly and applies the most appropriate interpretation via kriging with the associated accuracy and error analysis. IDW and splines provide an approximation of this but are not necessarily as accurate as kriging and do not provide the uncertainty information that kriging does.

Sub-bottom profiles were accessed from the CSIRO national geophysical survey and mapping data repository (www.cmar.csiro.au/data/gsm/search.cfm) to determine whether hard structures were buried under soft sediment. Single transects within Area 3 [34] and Area 4 [35] had been profiled using a Kongsberg TOPAS sub-bottom profiler. Sub-bottom profiles were not available for Areas 1, 2 and 5. Sediment samples were collected within each area using a Smith–McIntyre grab to quantify sediment composition within and among areas. Sub-samples from each grab, one each for grainsize, carbonate content (CaCO3%) and the identification of skeletal carbonate components, were analysed at Geoscience Australia [36, 37] (methods described in S1 Fig).

## Biological characteristics of the AC125

Benthic biodiversity was quantified along the AC125 using towed video and digital still imagery. Multibeam survey data guided the placement of towed video transects within each area–position combination (on, shallower, or deeper than the AC125 at the five study areas) following a generalized random tessellation stratified design [GRTS; 38]. This balanced design (transect numbers spread across a range of topographic complexity classes with spatial balance) guided transect start position and direction during field surveys, however, due to unfavourable environmental conditions (e.g., strong winds, currents, tides, low underwater visibility) not all transects were undertaken as planned. The towed video system was fitted with forward-pointing live video (for real-time navigation and broad habitat classification), a downward-pointing camera to capture high-resolution still images of the seafloor (for quantitative point sampling), light sources for both cameras and an ultra-short baseline (USBL) system for geolocating the towed body on the seafloor [39]. Most transects were between 1000–1500 m long and still images were captured every ~8 m along the transect from a height of 30–50 cm above the seafloor. Each high-resolution still image (Lumix LX100, 13 MP and Four Thirds sensor 17.3 x 13 mm, max resolution 4112 x 3088; field of view between ~ 0.1 and 0.25 m$^2$) was assigned a differential GPS position from the USBL. Depth was assigned to still images from multibeam survey data [28].

Downward-pointing still images were analysed using a point-intercept method: the benthos/substrate underlying five fixed spaced points per image was identified to the finest possible taxonomic classification or morphotype [40]. Unusable imagery was excluded from the benthic classification (i.e., images that were over-exposed, unfocussed, or obscured due to high sediment in the water column). An automatic image classifier platform (Benthobox software

[41]) was used to rapidly identify points overlaying sand/soft sediment, with all remaining points/images manually classified. The automatic image classifier accurately scores common homogenous categories (e.g., sand) but is less reliable at classifying lower abundance and more heterogenous categories such as sponges and other filter feeder biota. The CATAMI (collaborative and annotation tools for analysis of marine imagery) classification scheme was used to guide the categorisation of the benthos [42]. Identifications were aggregated into groups that were considered robust to observer variation, namely macroalgae, hard corals, other organisms, bryozoans, crinoids, hydroids, sponges (encrusting; other), whips, soft corals and gorgonians.

## Analyses of benthic data from still images

Transects with at least 30 images scored (regardless of transect length and spacing of images) were analysed to estimate benthic cover by area and transect. Multivariate analyses in PRIMER 7 [43] with PERMANOVA+ [44] were used to investigate benthic community patterns and whether latitude (five study areas) and/or position in relation to the AC125 (Shallow, AC125 or Deep) were important in structuring communities [44]. For transects that crossed position categories, imagery and data were split accordingly (e.g., Area 1, transect 1 was split into two components that represented AC125 and Deep). A minimum number of 10 images was required for inclusion in this analysis (~80 m of seafloor surveyed) to maximize replication within the 15 area–position combinations. Benthic groups (see S1 Table) selected for this comparison, excluded the group Microbenthos, as it largely comprised points with uncertain abiotic/biotic identifications (see Results section "Benthic habitats and communities of the AC125"). Percent cover data were square root transformed and Bray-Curtis similarity matrices were constructed. Hierarchical cluster analysis was carried out on pair-wise comparisons using group average clustering and the similarity profile analysis (SIMPROF test) was run for 999 permutations at a significance level of 5% to derive sets of groupings. A shade plot, combined with the cluster analysis, clarified the composition of benthic groups aggregating into SIMPROF groups. Non-metric multidimensional scaling (nMDS) ordinations were overlaid with SIMPROF groups to further explore patterns by study area and position. We tested for differences in benthic communities using PERMANOVA (999 permutations) in a hierarchical design with fixed factors for area and position. Type III sums of squares were used as they are more conservative for unbalanced designs [43]. The main PERMANOVA test was followed up with multiple sets of pairwise comparisons. Stacked bar graphs and density curves were created in R [45–47].

## Filling knowledge gaps with predictive modelling

Spatially continuous maps of predicted habitat classes for each of the five study areas were generated using the random forest (RF) ensemble machine-learning algorithm [48] in Python 2.7 (www.python.org). A global model was constructed using data from all five areas, which was then used to make predictions separately for each area. Forward-pointing live video taken during towed video surveys was assessed over 2 second intervals to define habitat classes (filter feeders, gorgonians, soft coral, sponge, whips, or no biota detected) and to classify their abundance (sparse, medium, or dense). These measures formed the dependent variables for the model. Fourteen secondary environmental parameters derived from the high-resolution (10 m pixel) bathymetry layer (Table 1) and shown from previous studies to be important in driving the distribution of benthic biota [49], were used as predictor variables in the model.

Thirty per cent of the habitat class and abundance data (from real-time video analysis) was withheld from building the model to enable model validation and estimates of predictive

**Table 1. Environmental parameters derived from high-resolution multibeam data used as independent variables in a random forest model to predict the spatial distribution of benthic habitat classes.**

| Description | Predictor |
|---|---|
| Depth | depth |
| Standard deviation of depth (5, 10, 25, 50-pixel window—corresponding to a circle of radius 50, 100, 250, or 500 m) | std_x |
| Range of depth (5, 10, 25, 50-pixel window—corresponding to a circle of radius 50, 100, 250, or 500 m) | rng_x |
| Slope | slope |
| Aspect | aspect |
| Plan curvature | plan |
| Profile curvature | prof |
| Overall curvature | curve |

accuracy [50, 51]. The overall classification accuracy of the model was estimated using the kappa statistic [52]; this statistic compares how well the model predicts each habitat class, how often the model finds a habitat class that was not observed, and how often this would be expected to occur by chance. Landis and Koch [53] rate the accuracy of the kappa statistic into five categories: slight (k = 0.01–0.2); fair (0.21–0.4); moderate (0.41–0.6); substantial (0.61–0.8) and almost perfect (0.81–1).

# Results

## Physical and biological characteristics of surveyed areas across the AC125

**Area 1.** Area 1 lies at the southern end of the AC125, closest to shore (18 km) and 235 km from Area 2 to the northeast. Shelf width was narrow and depths in Area 1 (70–181 m) indicate a moderately flat-topped platform in 70–80 m water and a gently inclined, west sloping seabed below 110 m (Fig 2). The platform and slightly inclined seabed were separated by an escarpment with slopes between 6–30˚ (Gently inclined to Steep; [54]) and the 125 m contour lies at the base of the platform and its escarpment. Seabed dunes were present at the foot of the escarpment, with an orientation suggesting current flow towards the south. Sediment samples (bivalve molluscs, benthic foraminifers) were deemed to be relict because the bioclastic carbonate was taphonomically degraded through fragmentation, dissolution, discolouration and recrystallisation from the original material [55, 56]. Bivalve molluscs in the gravel fraction (> 2 mm diameter) included *Barbatia* sp., *Plicatula* sp. and examples of the genus *Ostrea* and *Glycymeris*, all of which inhabit intertidal to subtidal waters [57, 58]. Across all five study areas, towed video analysis indicated average biotic cover was highest in Area 1 (1.07%, Fig 3), although this was attributable to a single transect that supported a diverse filter feeder community (S2 and S3B Figs). This transect was the shallowest across the five study areas (transect 3, depth 62–122 m) and had the highest biotic cover (11.2% cover c.f. second highest Area 2, 4.5% cover). It was the only transect in Area 1 to traverse the platform, with the remaining 11 transects (positioned on the AC125, or deeper) having low to nil cover (≤0.5%).

**Area 2.** Area 2 lies 145 km northwest of the mainland on a broad shelf, as do Areas 3–5. The AC125 was positioned in average depths of 135 m and dunes were present at about 150 m depth on the northwest facing slope (Fig 2). Just inshore of the AC125, the shallower, moderately flat to gently inclined seabed had in places, fine ridges consistent with coastal strandplain geometry (95–105 m depth). Seabed rugosity in the eastern parts in depths above the AC125, suggested subaerial exposure, erosion, and weathering. In contrast, the seabed below 150–160 m was smooth and most likely reflected the presence of muddy or fine sandy sediments. There was a trend across Area 2 for finer sediments to occur deeper than the AC125 and slightly

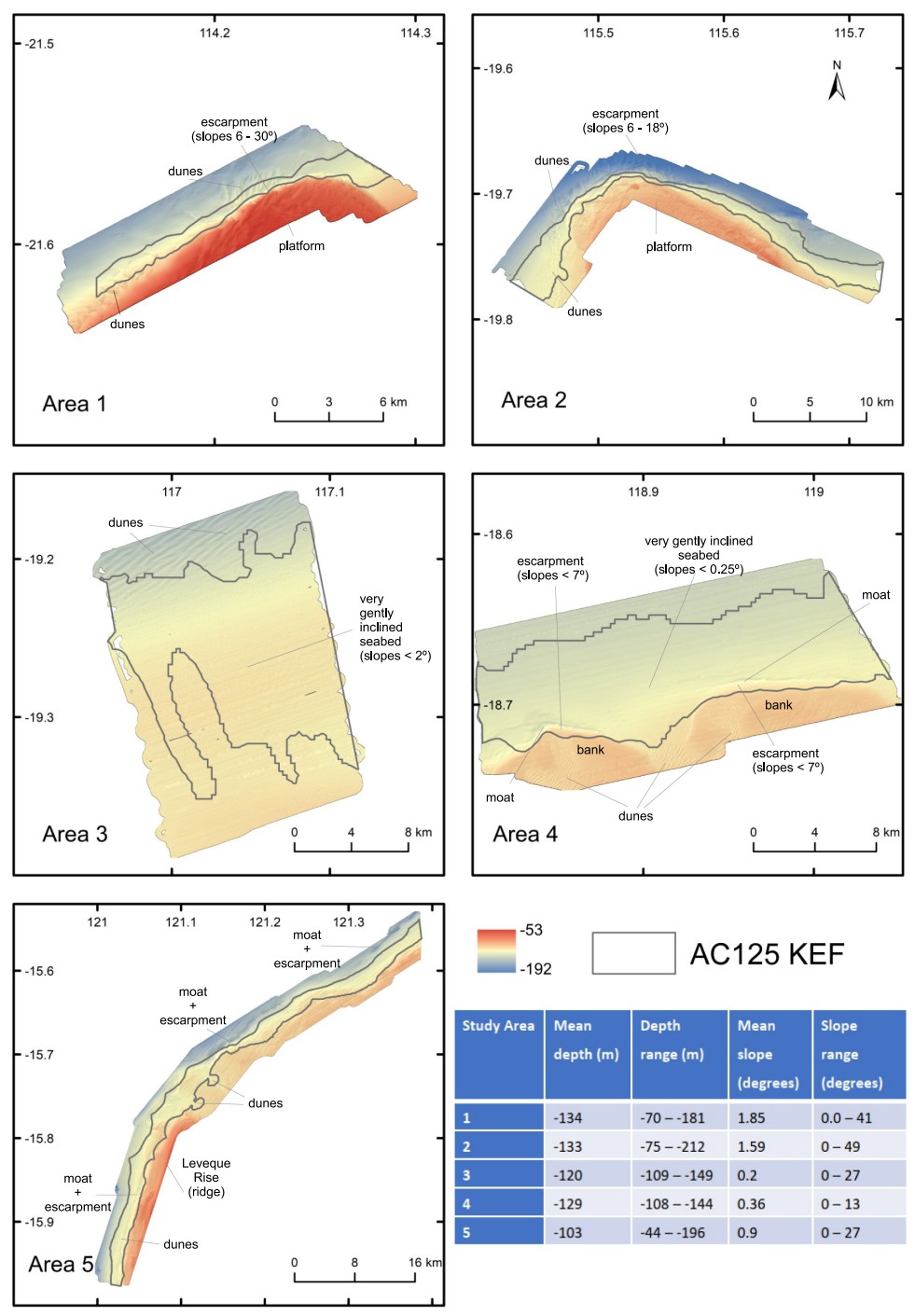

| Study Area | Mean depth (m) | Depth range (m) | Mean slope (degrees) | Slope range (degrees) |
|---|---|---|---|---|
| 1 | -134 | -70 – -181 | 1.85 | 0.0 – 41 |
| 2 | -133 | -75 – -212 | 1.59 | 0 – 49 |
| 3 | -120 | -109 – -149 | 0.2 | 0 – 27 |
| 4 | -129 | -108 – -144 | 0.36 | 0 – 13 |
| 5 | -103 | -44 – -196 | 0.9 | 0 – 27 |

**Fig 2. High-resolution (10 m) depth surfaces interpolated from multibeam bathymetry at five study areas.**
Horizontal scale is constant to illustrate relative depths. Notable features of each area are indicated.

coarser sediments to be present on the platform. Of the 24 towed video transects analysed, 16 recorded biota and average cover across Area 2 was 0.48%, the second highest after Area 1 (Fig 3). Gorgonians were the most abundant group and two transects (11 & 14) accounted for the

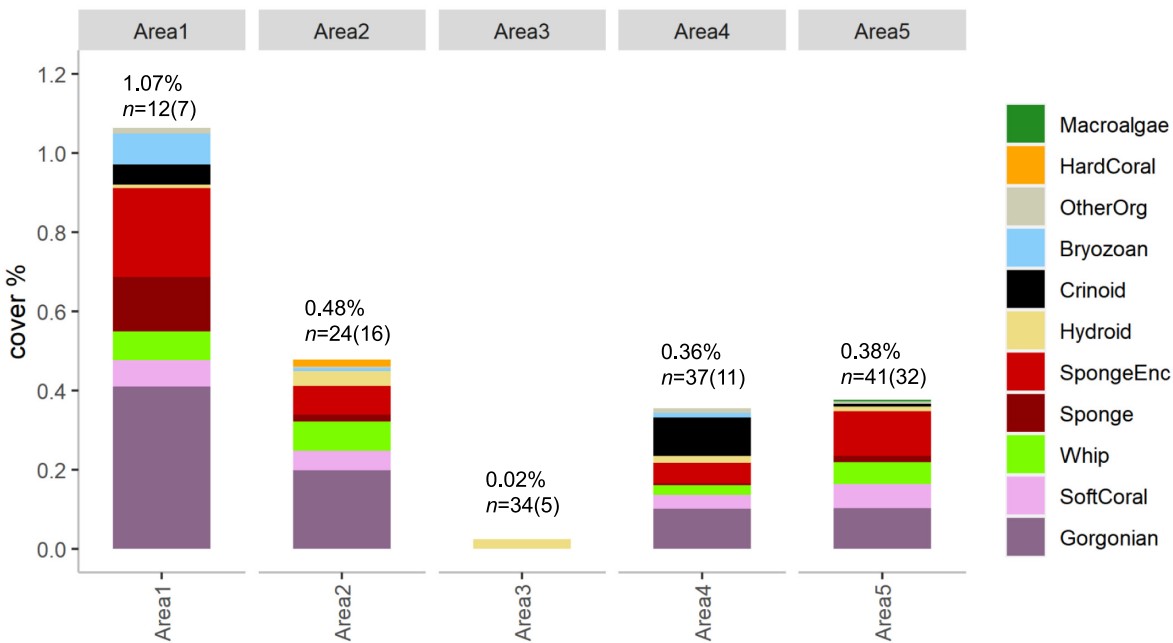

**Fig 3. Percentage cover of biotic groups at the five study areas.** For each area, labels indicate Total biotic cover (%), the number of transects (n) and of these the number of transects with biota recorded (bracketed number).

majority of biotic cover (3.1% and 4.5%, respectively) (S2 Fig; S1 Table). All other transects had biotic cover ≤ 0.8%.

**Area 3.** Area 3 lies approximately 145 km east of Area 2, and 130 km north of the mainland with depths between 109 m in the south and 149 m in the north. Hard surfaces were not obvious at the seabed, but many small shallow pits were present in the central and northern sections of Area 3. Discrete dunes were not observed, however across the northernmost margin, 3–4 m high (trough to crest) sinuous ridges were present with their axes aligned northeast-southwest (Fig 2). Overall, the shape of the seabed in Area 3 largely reflects the presence of unconsolidated sediments over an older structure. Sediment samples from the western and slightly deeper part of Area 3 were planktonic dominated with Pteropods and/or planktonic foraminifers comprising the largest proportion. Samples from the shallower, eastern side of Area 3 were dominated by either mollusc fragments or ooids. All samples were fine-grained either as mud or very fine sand. The presence of biota across this area was almost non-existent (0.02%), with only 5 of the 34 transects surveyed having any biota recorded at all (comprised solely of sparsely scattered hydroids) (S2 Fig, S1 Table).

**Area 4.** Area 4 lies 180 km east of Area 3 and 135 km north of the nearest coast with depths between 108 m in the south and 144 m in north, as in Area 3. To the north, the seabed was generally level to very slightly inclined with slopes less than 0.5°. To the south, lie two banks about 12–15 m higher than the intervening dune rippled seabed (Fig 2). Dunes were present on top of these banks as well as between, and moats located on the northeast side of the banks were aligned approximately orthogonal to the dunes. A single trochoidal dune present in the southernmost section of the bathymetry data had the dune crest aligned northeast–southwest as did most of the other dunes. The northern margins of the banks, and the moats to their north, had slopes of up to 5° and only rarely up to 13°. The northern margin of the western bank had a slight but noticeable diamond-shape in plan view. Shallow sub-bottom profile data across the westernmost end of Area 4, extending from area X5 of Jones et al. [59]

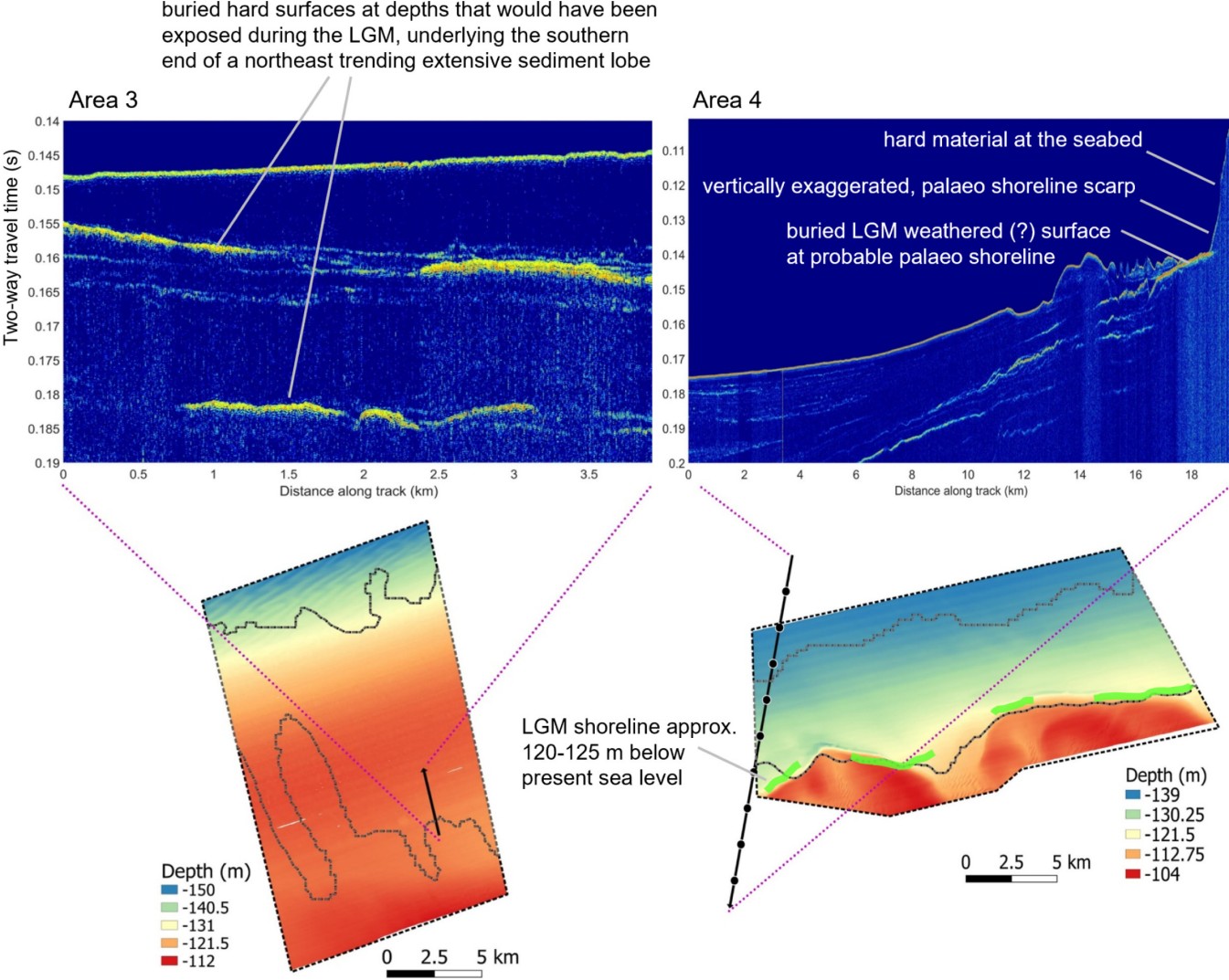

**Fig 4. Shallow sub-bottom profiles and seabed bathymetry at Area 3 and Area 4.** Buried hard surfaces are visible in yellow. The seabed in the centre of Area 3 is slightly higher than that immediately inshore and offshore and the sub-bottom profile suggests sediment depth is approximately 10 m. In Area 4, however, there is little sediment cover at approximately 120 m depth, and the seabed changes from a flat surface to slightly hilly [60]. If these 'hills' are not active sediment mounds, it is possible that this transition represents a palaeoshoreline (indicated by green line). The probable shoreline occurs at the same place as indicated in [61], profile F. The sub-bottom signal strongly suggests hard material is present at the shoreline scarp and therefore supports the idea of the presence of a previously subaerially exposed surface which could only have been exposed during the last glacial maximum (18.5–23 ka bp).

to the north of Area 4, and towards the inner shelf to the south, indicate the presence of shallow, north-dipping reflectors intersecting the seabed to the west of the western bank. To the south of Area 4, a single bank was visible in the sub-bottom data (Fig 4) with a moat present on its southern side, the opposite to the moats present to the north and northeast of the banks in Area 4. Masking of the sub-bottom profile signal show this bank had a hard feature. It was onlapped on its north and southern sides by less indurated sedimentary strata. Across Area 4, and in contrast to Area 3, the dominant sand fractions were benthic, either as ooids, or benthic foraminifers, or mollusc fragments. Mean grainsize indicated a predominance of sand sized sediment (0.063–2 mm diameter); coarser than for Area 3. Sediment types were predominantly of muddy sand, indicative of mixed material. Only 11 of the 37 transects in Area 4 had biotic cover and most of this occurs on transects shallower than the AC125 (on banks) or at

the junction with the AC125 (S2 Fig). Gorgonians and crinoids were the most dominant biotic groups in this area (Fig 3, S1 Table).

**Area 5.**    Area 5 extends across the western most portion of the Leveque Rise and onto the margin of the Leveque Shelf (Fig 2). It represents a distance of about 65 km along the AC125 and depths ranged from 44–196 m. Sloping areas of seabed, sediment dunes and moats were the key geomorphological elements present, the Leveque High notwithstanding. Slopes of up to 54˚ were present, but the majority of slopes were < 5˚, and in moats and their associated escarpments, slopes of up to 20˚ occurred. Dunes were present on the southern section in depths of 125–130 m. Rare trochoidal dunes, indicative of bidirectional and opposing currents (tidal), were also present. Sediment grainsizes were coarser than for the other study areas, tending towards coarse and very coarse sand sizes. Benthic sourced material was dominant in the sand and gravel size fractions. Benthic foraminifers were common to the west and south-west of the Leveque High, and samples to the north and northeast of the Leveque High were dominated by ooids. Area 5 had similar levels of biotic cover to Area 4 (0.38% and 0.36%, respectively) and encrusting sponges were the most abundant group followed closely by gorgonians (Fig 3, S1 Table). Biotic cover was recorded on 32 out of 41 transects, the highest proportion across the study, and transects supporting biota occurred in all positions (shallow, AC125, deep) (S2 Fig).

## Benthic habitats and communities of the AC125

From 148 towed video transects, benthic cover was estimated from 28,406 images comprising 139,075 points (points in images overlying benthos that could not be identified were omitted from the analysis) (S2 Table). Per study area, the number of transects ranged from 12 to 41, and mean transect depth ranged between 120 ± 0.05 (SE) m and 140 ± 0.12 m. Throughout the five areas, approximately 98% of total cover was soft sediment. Biota was not recorded along 77 of the 148 transects, and only 12 transects had biotic cover of > 1% (S2 Fig). Abiotic habitat types comprised soft sediments (mud/sand/silt), consolidated substrate (boulders/cobble/bedrock) or unconsolidated (rubble) substrate (S1 Table). The most common biota encountered were filter feeder communities, including gorgonians, sponges, soft coral and whip corals (Fig 3, S1 Table). The group Microbenthos encompassed points of uncertain abiotic/biotic identity (fine detrital material or small clumps on soft sediment; short branching to mat-like coverings on sand/gravel/rubble) (S1 Table; S3A Fig). Bryozoans were a component of some of these aggregations.

While soft sediment habitat was prevalent everywhere, consolidated substrate and benthic biota were largely absent in Area 3 (0.04 and 0.02%, respectively) (Fig 3; S1 Table). Biota were patchily distributed, and the highest average biotic cover occurred at Area 1 (1.07%), although this was attributable to a single shallow transect (transect 3) which had 11.2% cover (S2 Fig). Soft coral (gorgonians, sea whips, and all other soft corals) and sponge groups accounted for most of the biotic cover across Areas 1, 2, 4 and 5, while other biota (e.g., bryozoans, other organisms) although present across most areas, were less well represented. Gorgonians were the most abundant biotic group, followed by encrusting sponges, all other soft coral, sea whips, crinoids, and hydroids (Fig 3; S1 Table).

Gorgonians, crinoids, and hydroids had clumped depth distributions in some study areas, whereas other biota, such as sponges, whips and soft coral were more evenly distributed (Fig 5). Crinoids (S3C Fig) generally occurred deeper than other biota, as did hard coral, which was only recorded in Area 2 on transect 9 at depths of 152–154 m. This coral (S3D Fig) resembled *Desmophyllum pertusum* (Family Caryophylliidae), a deep-water azooxanthellate scleractinian coral. Hydroids predominantly occurred deeper than 100 m.

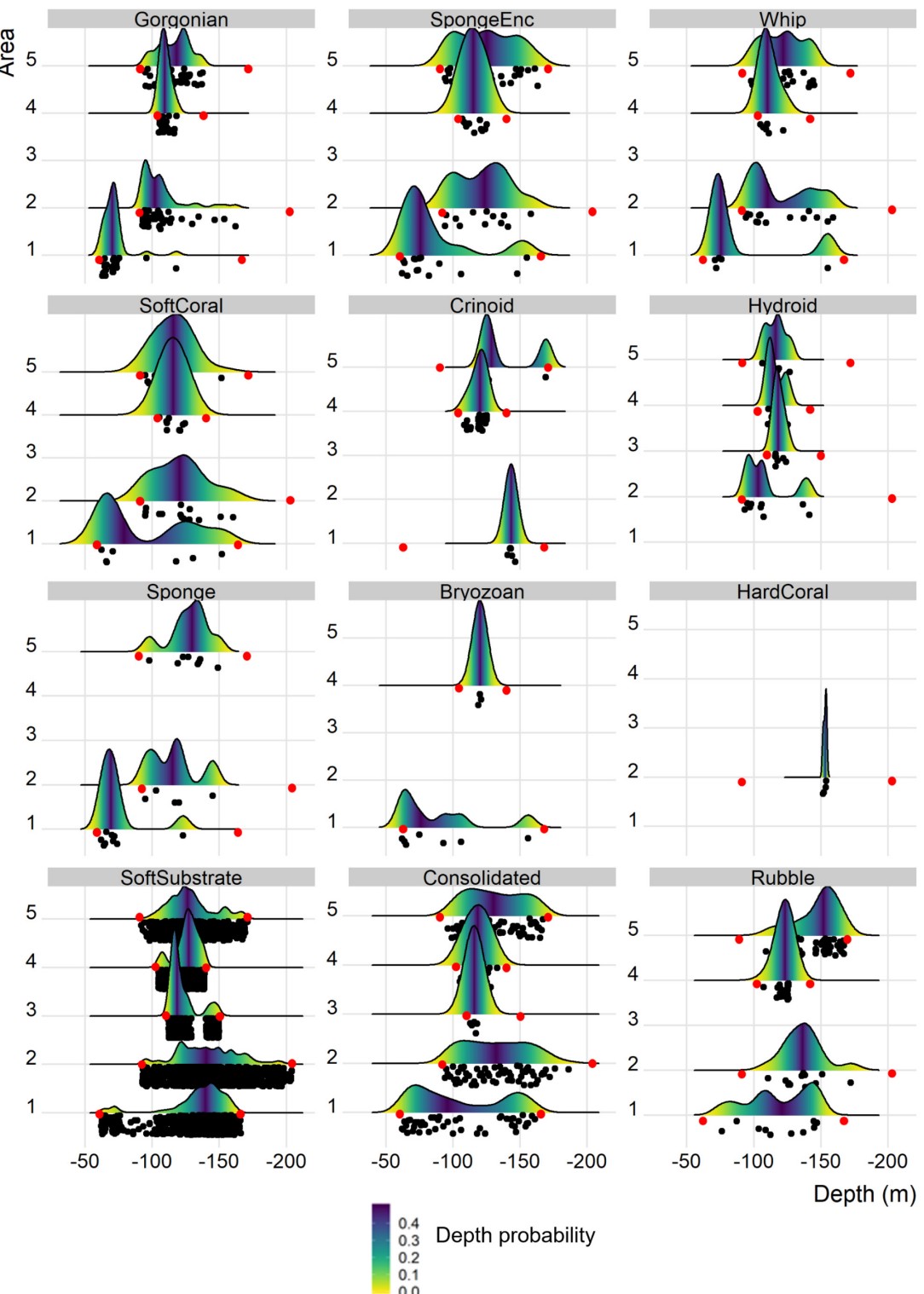

**Fig 5. Density curves of biotic groups (rows 1–3) and substrate types (row 4) at each area ordered by prevalence across the study.** Black dots beneath each curve represent samples (points scored from images) at each area. Groups with restricted depth distributions (e.g., gorgonians, crinoids and hydroids) are indicated by black dots clumped rather than spread across the sampled depth range. Red dots indicate the minimum and maximum depth of total sampling at each area. Density curves are coloured by depth probability (the empirical cumulative density function for the distribution).

**Spatial patterns.** There was no evidence of a latitudinal gradient in benthic community composition or substrates along the AC125, despite original suggestions of a seascape change around 17˚S associated with primary productivity [22]. The 15 area–position combinations fall into five groups, broadly defined by low biotic cover (cluster a) or an increasing dominance of biotic cover and consolidated substrate and rubble (Fig 6). Shallow positions typically had higher proportions of consolidated substrate than areas on the AC125 or deeper than the AC125 (Fig 6).

There were significant differences in benthic communities among the study areas and positions relative to the AC125, including an interactive effect of area x position on benthic habitat structure (Table 2). Pairwise analyses showed that benthic composition within areas 1, 3 and 5 was consistent across positions (shallow, AC125 or deep; S4 Fig). The shallow position in Area 2 was significantly different to AC125 and deep, which were similar. In Area 4, there was a significant difference in benthic composition between shallow and deep positions. Within positions, and specific to shallow, benthic composition of Area 3 was significantly different from Areas 1, 2 and 4, but the same as in Area 5. Area 3 was different from Areas 2 and 5 within the AC125, while only Areas 2 and 4 were different in the deep position.

**Habitat mapping.** Eleven benthic habitat classes were observed across the five areas based on 101,634 scored real-time towed video records (S3 Table). The three most common classes were No biota detected (65,358), Filter feeders–sparse (30,051) and Filter feeders–medium (5,936). Filter feeders represent a mixed community, comprising gorgonians, soft coral, sponge, and whips. Separate categories were also used for each of these biotas where they existed in non-mixed communities. Not all habitat classes were observed in each area.

Based on a global model that used all training data across the five study areas and the full suite of environmental predictors (Table 1), depth was the most important predictor of benthic habitats (19%) followed by depth range at the broad scale (rng50; 14.8%), and the standard deviation of depth at the broad scale (std50; 12.3%). This indicates that depth and broad-scale structural complexity of the seafloor were the best surrogates for predicting habitat class (S5 Fig).

Only three of the habitat classes were common; the No biota class was predicted to cover the largest proportion of each area, however this varied from > 99% in Area 3 to < 50% in Areas 1, 2 and 5 (Fig 7, S4 Table). Filter feeders–sparse were more prevalent than Filter feeders–medium (28.6% versus 6.8%). Of the latter, about twice as much was predicted for Area 1 (24.05%) compared to the other areas (Fig 7; S4 Table). The class Other Biota, which comprised the remaining seven habitat classes, were rarely predicted to occur across the study areas (Fig 7; S4 Table).

The predicted distributions of Filter feeder habitats, both -sparse and -medium, were highest in the shallow position, followed by on the AC125 and then the deep position (Fig 8). The exception was Area 5 where Filter feeders–sparse were predicted to be more prevalent on the AC125 than the shallow position. Filter feeders—medium were always more prevalent in the shallow position unless completely absent (as in Area 3). Soft sediment areas with No biota detected were predicted to be most prevalent deeper than the AC125 in Areas 1 and 2, and similar to the AC125 in Areas 3 and 5. In Area 4, Filter feeder habitat was only present in the southern one-third of the study area, primarily in the shallow position, followed by on the AC125.

The RF model was generally accurate in predicting the observed benthic habitat classes in the field (Fig 9). Misclassification mostly occurred between the No biota detected and Filter feeders–sparse classes. The model was more likely to overpredict than underpredict No biota detected. In contrast, the model was more likely to underpredict Filter feeders–sparse. Calculated classification accuracies (as measured by kappa; S4 Table) for each area were consistently high (almost perfect, kappa ≥ 0.91) for all but Area 3 (kappa = 0.76).

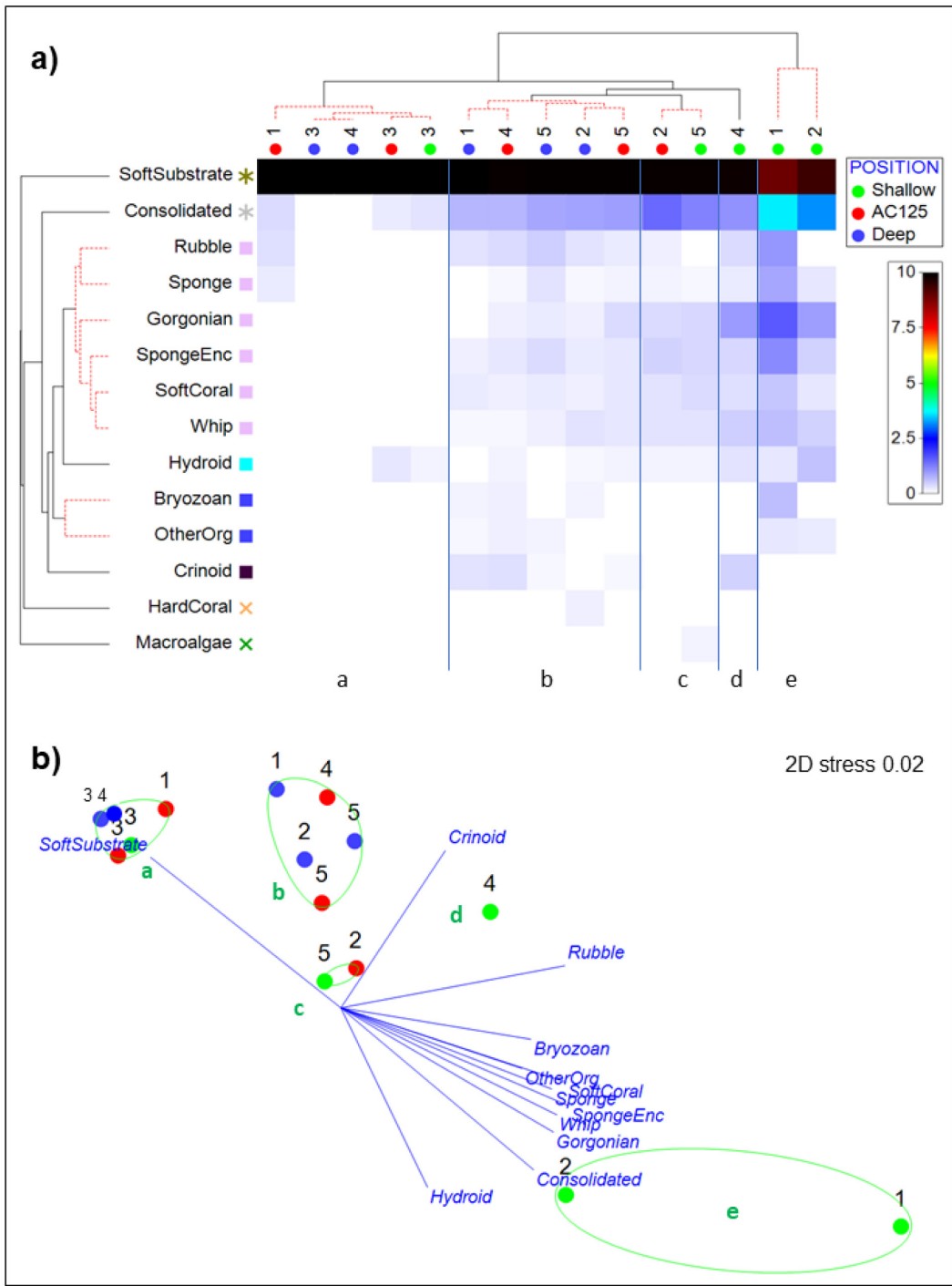

**Fig 6. a) Shade plot coupled with cluster analyses.** The abundance of benthic groups (y axis) by area–position combinations (x axis) are indicated by the linear intensity scale (proportional to square-root percent cover of the benthic groups). Area–position combinations are grouped according to the five SIMPROF groups (a–e). **b) Two-dimensional representation of a non-metric multidimensional scaling (nMDS) ordination of area–position combinations based on the cover of benthic groups.** The plot is overlaid with SIMPROF groups (a-e) from the cluster analysis and vectors for benthic groups with correlation >0.5.

**Table 2. Results from PERMANOVA on fine-scale benthic categories.** The model was constructed with Bray–Curtis similarities with area and position fixed factors. df: degrees of freedom, SS: sums of squares, MS: mean square, Pseudo-F:permuted F-statistic, P(perm): permuted probability, perms: number of permutations. Values in bold represent significant results.

| Source | df | SS | MS | Pseudo-F | P(perm) | Unique perms |
|---|---|---|---|---|---|---|
| Area | 4 | 1837.1 | 459.27 | 6.6489 | **0.001** | 999 |
| Position | 2 | 1662.3 | 831.13 | 12.032 | **0.001** | 996 |
| Area x position | 8 | 1549.3 | 193.66 | 2.8037 | **0.010** | 999 |
| Residuals | 175 | 12088 | 69.08 | | | |
| Total | 189 | 15653 | | | | |

## Discussion

The new bathymetry, sediment and benthic habitat data provide critical and novel insights about the seabed and biota of the northwest shelf of Australia. Study areas have water depths that suggest at least part of the seabed may have been in a coastal setting during the LGM [19]. However, we did not find direct geomorphological evidence for the existence of a palaeoshoreline formed during the LGM preserved across the AC125, though there were some indications of candidate areas to find palaeoshoreline material at or just below the modern seabed. Furthermore, the new bathymetry and sediment data enables some empirical revision of the hypothesis that steep escarpments and submergent shorelines are present on the northwest shelf [20, 61]. Accordingly, a distinct zone of benthic habitat and biota typically associated with hard substrate anticipated to be part of an ancient coastline, was not observed in this study. Instead, most study areas were characterised by expanses of soft sediment seabed with negligible biota, interspersed by pockets of hard substrate supporting filter feeder communities. A key recommendation is for the Australian Government to reconsider whether this area deserves KEF status in future revisions of the offshore bioregional areas.

### Geomorphological indications of a palaeoshoreline

While we did not find exposed hard substrate representing a drowned ancient coastline, there was considerable evidence pointing to the existence of a buried shoreline across at least parts of the study area. Sediment samples across the study areas were composed of primarily 'old', degraded sub-fossil material and smaller proportions of recently dead (taphonomically 'young') biogenic carbonate, implying the ongoing reworking of sediment across this shelf (S1 Fig). In Area 1, disarticulated shallow marine (littoral) bivalves, *Plicatula* sp., and *Barbatia* sp. recovered from the seabed were indicative of an intertidal to subtidal setting [62–69] despite being recovered from depths of 120–130 m. Their recovery in muddy brown sediment at a depth expected for the LGM sea level suggest a preserved lowstand shoreline may exist at these locations, or nearby (S1 Fig). Similarly, preserved shoreline deposits may be buried beneath modern seabed sediment in Areas 3 and 4, however whether a palaeoshoreline exists will not be completely resolved without seabed coring and more detailed bathymetric surveys (S1 Fig). In Area 2, the coastal strandplain-like geometry of the platform occurs 100 m or so below sea level and most likely developed approximately 15 ka b.p., after the LGM, when sea levels were higher than during the last ice age [70]. Much less likely, the strandplain could have formed during an earlier interval of low sea level. Sub-bottom data examined for Areas 3 and 4, indicate hard surfaces buried beneath the modern seabed (Fig 3). None of these buried hard surfaces intersect the seabed in Areas 3 or 4, however intersection with the seabed does occur some 2–4 km southwest of Area 4 in depths of 110–115 m. Rock, if present at those depths during the LGM, would have been above sea level and exposed to terrestrial conditions. Some of

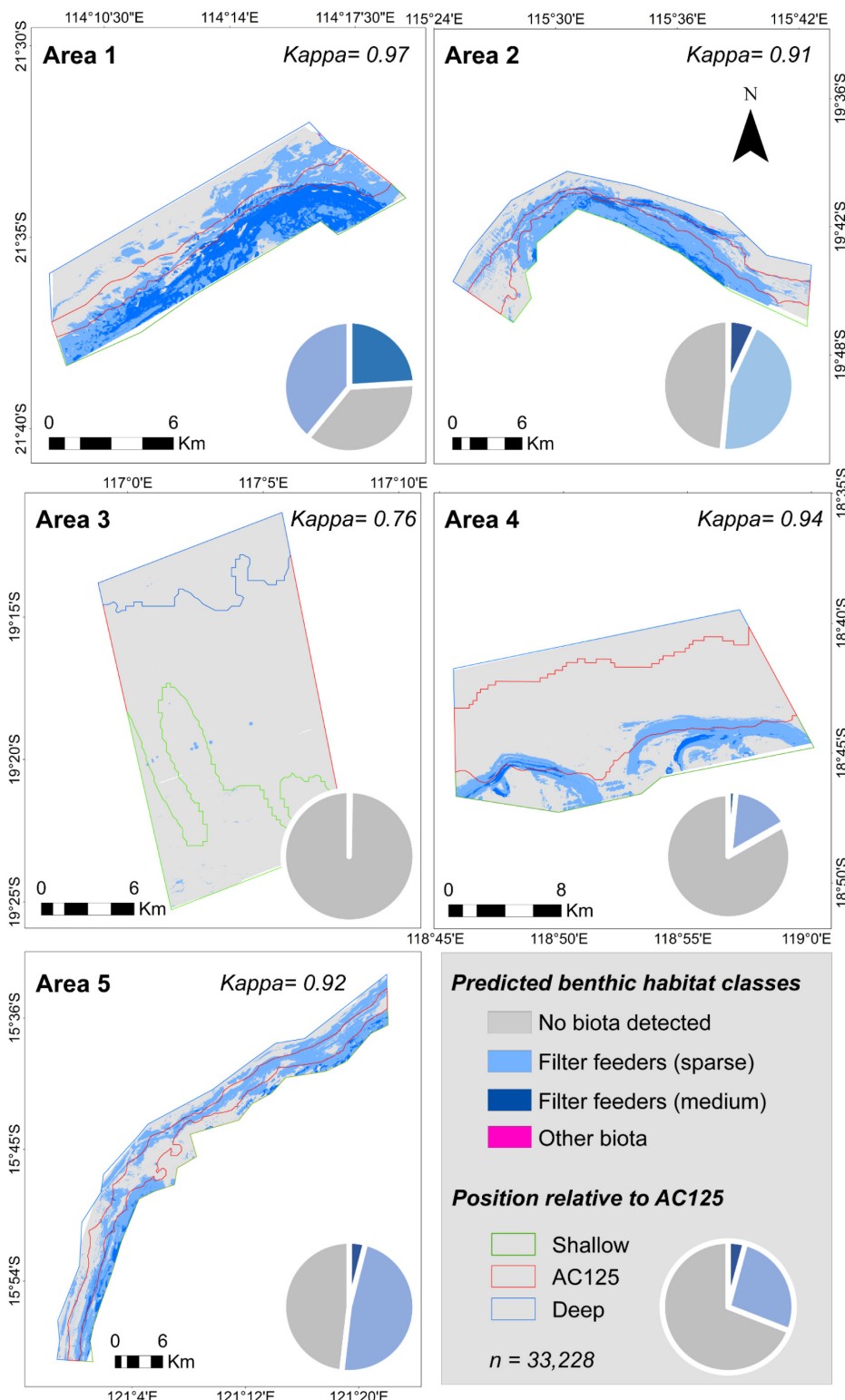

**Fig 7. Predicted spatial distribution of the three dominant habitat classes (no biota detected, Filter feeders— sparse, Filter feeders—medium) and the combined class other biota (comprising the seven remaining classes) at the five study areas.** Other biota is rarely predicted and difficult to identify on the maps and does not rate on the pie graphs. Note that each area includes seafloor deeper than the AC125 (blue line), on the AC125 (red line) and shallower

than the AC125 (green line). Predictions are based on 33,228 towed video observations of habitat, divided into a training (70%) and testing (30%) set. Pie charts indicate the prevalence of each habitat class in each area and for all areas combined (bottom right). Classification accuracy, as measured by kappa, was very high in each area.

the hard strata that underlie Areas 3 and 4 may have been exposed when sea levels were at LGM values (-125 m). However, it was not possible to tie specific buried horizons to the LGM sea level without coring and dating, both beyond the remit of this study. In Area 4, the flat sea-bed intersects with the gently inclined escarpments at the periphery of the banks in depths of 125–130 m. The presence of escarpments supports the hypothesis for the existence of a palaeoshoreline across Area 4 as suggested by James et al. [20] because the base of the escarpments occur in depths close to that determined for Australian sea level during the LGM [70–72]. James et al. profile F, has a near vertical cliff as the profile crosses Area 4 [20]. The new bathymetry indicates depths of 110–115 m at this location, slightly deeper than their Profile F indicates (100 m) and significantly shallower slopes. The seabed profiles of Jones [73] and James et al. [20] were vertically exaggerated to illustrate major changes in profile shape across the shelf. Unfortunately, the vertical apparent cliffed escarpments indicated for profile F do not exist in high resolution bathymetry (this study), with escarpment slopes of only 5° present in Area 4. Moats and escarpments were also key features in Area 5, existing at the junction of shelf and deeper water geological sub-basins that comprise the Browse Basin. In profile, the moats tend to have u- to v-shaped morphologies, suggestive of an absence of fill despite the strong currents and available sediment. The presence of benthic biota in Areas 4 and 5 as determined by towed video, suggests moat and escarpment surfaces were largely hard rather than soft. Immediately inshore of Area 5 the seabed was predominantly hard overlain by a veneer of carbonate sediments [36, 74].

## Sedimentary processes along the AC125

Approximately 98% of the seabed surveyed was comprised of unconsolidated soft sediment habitat (mud/sand/silt), which suggests that post-glacial sediments have infilled parts of the AC125. This sediment infill was not unexpected given the AC125 falls within a section of the Australian continental shelf where tidal currents frequently generate enough bottom shear stress to move sediments, i.e., highly energetic nearbed flows result from the internal waves generated by large tides [75, 76]. Sediment grain sizes tend to be dominated by skeletal

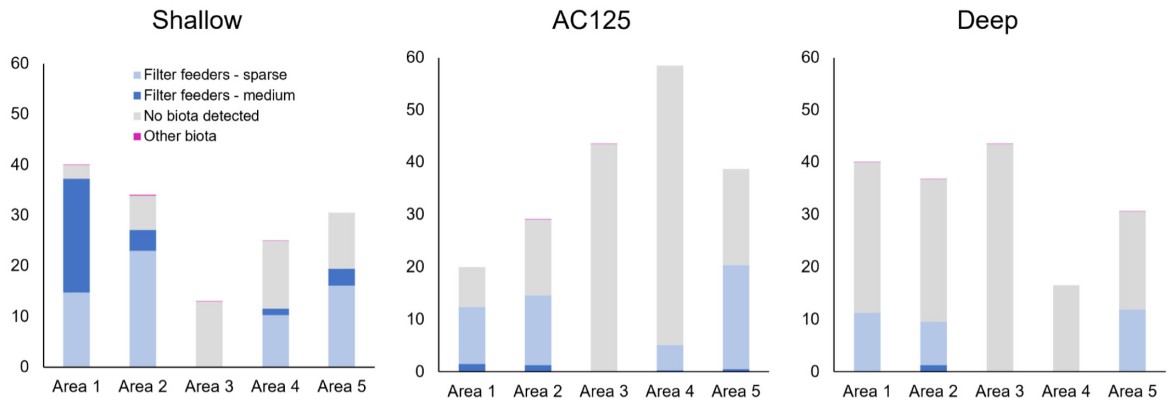

**Fig 8. Percentage area of predicted habitat classes across the five study areas grouped by position in relation to the AC125 (shallow, AC125, deep).** Percentages are calculated as the percentage of all habitat located within each area.

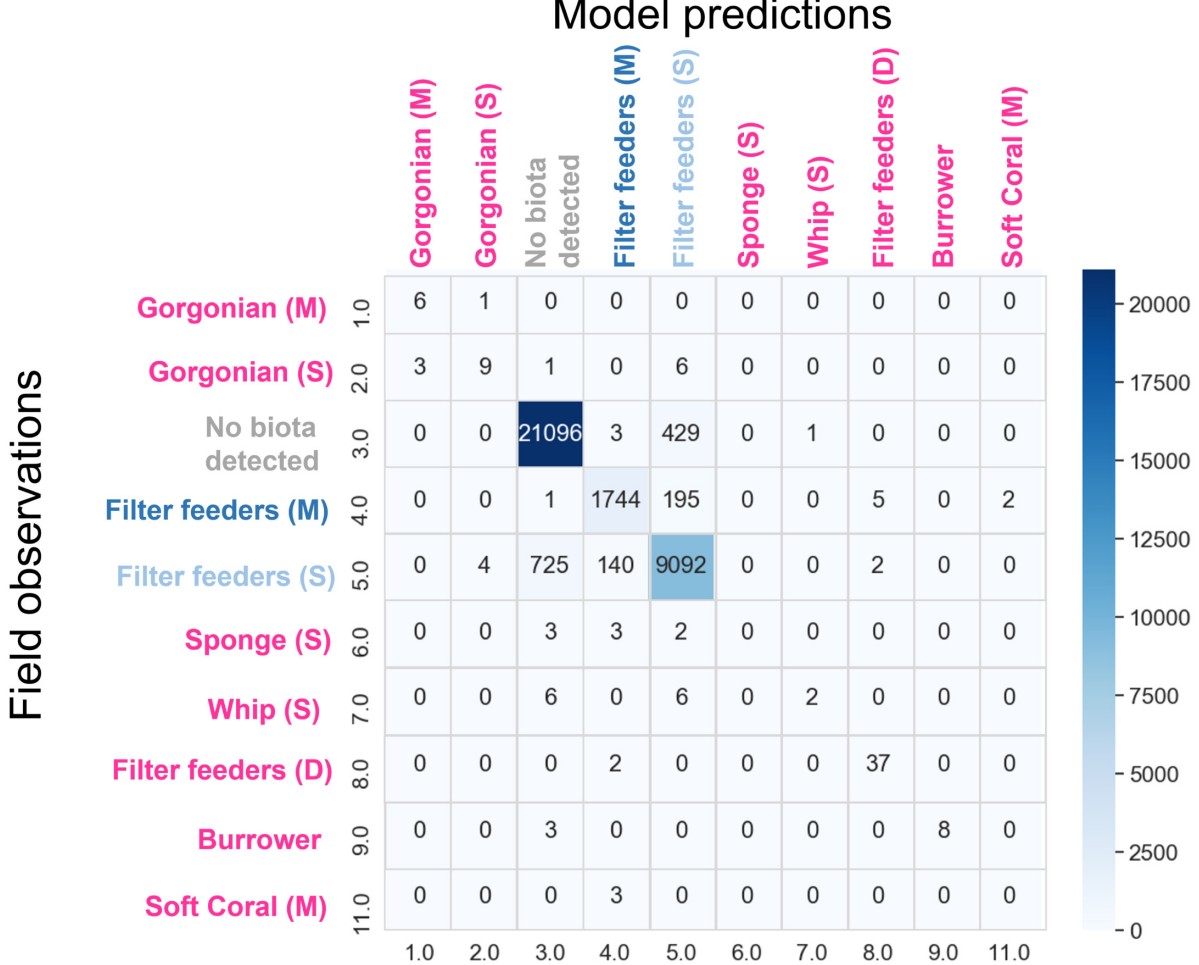

**Fig 9. Confusion matrix for the spatial benthic habitat model.** The y axis indicates the classes observed in the field, and the x axis shows the classes predicted by the model. Numbers along the diagonal indicate where the model predicted correctly. Numbers off the diagonal indicate where the model predicted incorrectly. Habitat types are colour coded to match Fig 7.

carbonate sand; its low-density making transport easier [77]. Frequent cyclone activity in the region [78] can also resuspend and transport bottom sediments, particularly when storms propagate along-shore [79]. The principal sedimentological features observed were current-generated submarine dunes, present in discrete dunefields in all the studied areas. Dune crests were generally oriented north to northeastwards suggesting the formative process may be similar. The southern most areas, Areas 1 and 2, and partly 3 are different from 4 and 5 because they are directly impacted by the Leeuwin/Holloway current; a shelf-edge oceanic current, distinct from the inshore currents, across-shelf internal waves, and tidal currents. The Leeuwin/ Holloway current flows seasonally, and spatially, at the water depths surveyed and would disallow to some extent modern sediment settling at those survey areas [80–82]. The presence of the current, and the absence of other major water body influences on the southernmost areas may potentially lead to habitat differences between these areas and the more northern survey areas. Rare trochoidal dunes were observed in Areas 4 and 5, indicating that for the northern section of the AC125 examined here, bidirectional, cross-shelf, probably tidal currents are responsible for some sediment mobilisation. The combination of planktonic skeletal carbonate being more common in the southern study areas (1, 2 and 3) over the northern areas and the

presence of trochoidal dunes in Areas 4 and 5 but not in 1, 2 or 3, and recent observations on the Holloway current, suggests that the southern study areas are more influenced by oceanic conditions than Areas 4 or 5. The thickness of soft sediment present at the seabed (the primary geological sedimentary stratum, and the soft substratum for benthic habitat mapping) seemed to increase with increasing water depths below and west of the AC125 (this study; see also [59]).

## Benthic habitats and communities of the AC125

Biological communities across the study area were sparse and patchy, largely due to the high proportion of unconsolidated soft sediment habitat (mud/sand/silt) which supported negligible epibenthic biota. Hard substrate is obligatory for the recruitment of sessile benthic invertebrates, with fine-scale topography on surfaces providing larval attachment points [83, 84]. New recruits were difficult to detect in imagery however it was unsurprising that the highest percentage cover of benthic biota typically occurred where the seabed was more rugose (S2 Fig).

Observations from the towed video data did not identify a distinct zone of benthic habitat along the AC125 indicative of a hard substrate ancient coastline. In general, the AC125 was characterised by soft substrate interspersed by patches of hard substrate supporting filter feeder communities. Areas shallower than the AC125 typically had greater cover and higher diversity of biota, while on the AC125 and deeper, benthic communities were typically sparse and low in diversity. This aligns with bathymetry data (1:10,000 to 1:250,000) which shows the seabed in water shallower than the AC125 appears more rugose than where finer and commonly muddy sediment occurs in deeper water. The distribution of benthic biota and habitats also mirror those of Currey-Randall et al. [85], which indicate highest fish diversity in areas shallower than the AC125, and only sections of the AC125 with hard bottom substrate supporting enhanced fish diversity. Spatial models generated for each study area show patterns in predicted benthic habitat classes to broadly align with observed distributions of benthic groups (Fig 7) from towed video transects. The predictive maps also show that apart from one study area (Area 5), filter feeder habitat classes were always more common in waters adjacent to, but shallower than, the AC125 (Fig 8).

The lack of hard substrate likely explains why the benthic habitats formed along the AC125 were patchily distributed and sparse. Another study near Lynher Bank on the northwest shelf (shallower than AC 125; 30–104 m depth, average 63 m depth) also had low biotic cover similar to that occurring along the AC125 (average 6% across all transects c.f. average < 1% across all our AC125 study areas) [86]. Even if sparse, these patchy benthic communities are important as they provide habitat structure for a range of invertebrates [87, 88] and are also likely to be important for demersal fish species on the AC125 [36]. Deeper parts of reef/shoal features occurring near to the AC125 (i.e., Area 1: Ningaloo Reef and Muiron Islands; Area 2: Rankin Bank; Area 3: Glomar Shoal) that support mesophotic biota [28–30] may be important sources of recruitment for the AC125. For example, in Area 1, the platform which lies shallower (70–80 m depth) than the AC125, appears to be predominantly hard with potential to support substantial filter feeder communities. However additional surveys are required to determine the extent and complexity of habitat on this platform, and whether connectivity exists between benthic communities at this location and nearby Ningaloo Reef and the Muiron Islands (within ~18 km), both of which are known hotspots for sponge and filter feeder communities [29, 30]. Differences in community composition between the AC125 and adjacent areas, as well as differences associated with longitudinal gradients, may be apparent at the species level, however more detailed imagery and specimen collections would be required to provide

species-level resolution. In addition, it is recognised that quantifying cover of sparse filter feeder communities from downward-pointing imagery underestimates cover, as filter feeder communities project most of their biomass upward into the water column, rather than spreading horizontally across the seafloor as in coral reefs. Estimates of biotic cover in these habitats could be improved by adjusting sampling methodology to include more points per image, by estimating cover from an oblique perspective (rather than top-down), or by identifying all individuals in an image and assigning size classes.

## Is the AC125 really a key ecological feature?

The AC125 was designated as a Key Ecological Feature because it coincides with predicted sea-level prior approximately 17,000 years ago and as such was expected to provide areas of hard substrate that would support high diversity and species richness relative to surrounding soft sediment areas (e.g., [24]). Here, and through associated studies [85] we have shown that, if a distinct coastline exists, it is now largely buried and as such does not provide a unique hard substrate habitat. Rather, the communities of the AC125 are representative of the lower-mesophotic zone of the northwest shelf, comprising a mix of hard- and soft-sediment communities along its length, and with the most diverse and abundant marine communities typically shallower than 125m.

However, much work remains to fully characterise the biota along the entire extent of the AC125. Sub-bottom profiling offers a mechanism for detecting buried hard substrate that could help map the location of the ancient coastline within the AC125 when it is not observable using multibeam. Such profiles could guide further towed video surveys and more comprehensive data that would help target and catalogue the biotic diversity of the vast region of the AC125. Coupling this data with coring and additional multibeam surveys in shallow areas adjacent to the AC125 may identify whether some sections of the ancient coastline lie outside the currently defined KEF.

Additionally, oceanographic studies could also inform decisions about the ecological value of the AC125, especially in the context of upwelling, vertical mixing and tidal fronts in the region. This knowledge will inform benthic biodiversity studies, but also contribute to understanding whether the AC125 is important for productivity and/or an important migratory route for megafauna. For example, flatback turtles in NW Australia may use ancient coastlines to aid navigation either directly linked to water depth or associated tidal fronts [89], although the relationships between the AC125 and other migratory species is currently less clear (e.g., [90]).

Our study is one of the first to extensively survey one of the largest KEFs in Australia and this baseline knowledge is crucial to inform effective management of biodiversity in Australia's marine estate. On the northwest shelf in particular, increasing pressures from industry (including fishing, oil and gas exploration and development, shipping, and tourism) provide an impetus for evidence-based decision-making. The improved knowledge of the AC125 gained here provides an important first step towards implementing management strategies and identifying gaps in our knowledge concerning the bathymetry and biodiversity in these regions.

## Supporting information

**S1 Fig. Methods description for sediment analysis undertaken by Geoscience Australia.** (PDF)

**S2 Fig. Towed video transects at the five study areas.** Maps show the AC125 (blue) and areas shallower (light gray) and deeper (dark gray) than the AC125. Transect lines are coloured to

indicate benthic composition and bar plots show the biotic composition of transects. n = the total number of transects surveyed in each area. Only transects with biota are shown in bar graphs.
(PDF)

**S3 Fig.** Downward-pointing still imagery from towed video surveys showing a) example of microbenthos, b) filter feeder community, c) crinoids and d) hard coral colony.
(PDF)

**S4 Fig. Significant pairwise tests within the interactive term 'area x position' are coloured red (P ≤ 0.05), with non-significant tests coloured green.**
(PDF)

**S5 Fig. Relative importance of predictor variables in the global benthic habitat model fit to the five areas (refer to Table 1 for explanation of variables).**
(PDF)

**S1 Table. Percentage cover of benthic groups at the five study areas.**
(PDF)

**S2 Table. Towed video transect data summarised by study area.**
(PDF)

**S3 Table. Testing and training data distribution by study area and habitat class.**
(PDF)

**S4 Table. Percentage of three dominant habitat classes (plus a mixed class: Other biota) in each of five study areas.**
(PDF)

## Acknowledgments

AIMS' North West Shoals to Shore Research Program was proudly supported by Santos as part of the company's commitment to better understand Western Australia's marine environment. The industry partners had no role in the data analysis, data interpretation, the decision to publish or the preparation of the manuscript. AIMS acknowledges the Yinggarda, Baiyungu, Thalanyji, Yaburara, Mardudhunera, Ngarluma, Yindjibarndi, Ngarla, Bindunbur and Bardi Jawi People, as Traditional Owners of the Country adjacent to the ancient coastline where this work was undertaken. We recognise these People's ongoing spiritual and physical connection to Country and pay our respects to Aboriginal Elders past, present and emerging. We also thank all the participants in fieldwork including the masters and crew of the RV *Solander*, Marcus Stowar, Matt Birt, Kathy Cure, Scott Gardner, Shaun Hahn, Simon Harries, Anne Kennedy, Nick Logan, and Sarah-Jayne Pyke.

## Author Contributions

**Conceptualization:** Mary Wakeford, Marji Puotinen, William Nicholas, Jamie Colquhoun, Steve Whalan, Iain Parnum, Ben Radford, Ronen Galaiduk, Karen J. Miller.

**Data curation:** Mary Wakeford, Marji Puotinen, William Nicholas, Mark Case.

**Formal analysis:** Mary Wakeford, Marji Puotinen, William Nicholas, Iain Parnum, Ben Radford.

**Funding acquisition:** Karen J. Miller.

**Investigation:** Mary Wakeford.

**Methodology:** Mary Wakeford, Marji Puotinen, William Nicholas, Jamie Colquhoun, Iain Parnum.

**Writing – original draft:** Mary Wakeford, Marji Puotinen, William Nicholas, Brigit I. Vaughan, Steve Whalan, Iain Parnum, Ronen Galaiduk.

**Writing – review & editing:** Mary Wakeford, Marji Puotinen, William Nicholas, Jamie Colquhoun, Brigit I. Vaughan, Steve Whalan, Iain Parnum, Ben Radford, Ronen Galaiduk, Karen J. Miller.

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
