## [Decision Letter · Decision Letter 0]

10 May 2023

PONE-D-23-07404Mesophotic benthic communities associated with a submerged palaeoshoreline in Western AustraliaPLOS ONE

Dear Dr. Wakeford,

Thank you for submitting your manuscript to PLOS ONE. After careful consideration, we feel that it has merit but does not fully meet PLOS ONE’s publication criteria as it currently stands. Therefore, we invite you to submit a revised version of the manuscript that addresses the points raised during the review process.

We look forward to receiving your revised manuscript.

Kind regards,

Shamim Ahmad, PhD

Academic Editor

PLOS ONE

Journal Requirements:

"AIMS’ North West Shoals to Shore Research Program is proudly supported by Santos (www.santos.com) as part of the company’s commitment to better understand Western Australia’s marine environment.  The industry partner had no role in the study design, data collection and analysis, the decision to publish or the preparation of the manuscript."

"AIMS’ North West Shoals to Shore Research Program was proudly supported by Santos as part of the company’s commitment to better understand Western Australia’s marine environment. The industry partners had no role in the data analysis, data interpretation, the decision to publish or the preparation of the manuscript. AIMS acknowledges the Yinggarda, Baiyungu, Thalanyji, Yaburara, Mardudhunera, Ngarluma, Yindjibarndi, Ngarla, Bindunbur and Bardi Jawi People, as Traditional Owners of the Country adjacent to the ancient coastline where this work was undertaken. We recognise these People’s ongoing spiritual and physical connection to Country and pay our respects to Aboriginal Elders past, present and emerging. We also thank all the participants in fieldwork including the masters and crew of the RV Solander, Marcus Stowar, Matt Birt, Kathy Cure, Scott Gardner, Shaun Hahn, Simon Harries, Anne Kennedy, Nick Logan, and Sarah-Jayne Pyke."

"AIMS’ North West Shoals to Shore Research Program is proudly supported by Santos (www.santos.com) as part of the company’s commitment to better understand Western Australia’s marine environment.  The industry partner had no role in the study design, data collection and analysis, the decision to publish or the preparation of the manuscript."

"Santos is the commercial funder of this research yet they had no role in the study design, data collection and analysis, the decision to publish or the preparation of the manuscript. The funder does not alter our adherence to PLOS ONE policies on sharing data and materials."

6. Please amend the manuscript submission data (via Edit Submission) to include author Dr. Steve Whalan.

7. We note that Figure 1 in your submission contain [map/satellite] images which may be copyrighted. All PLOS content is published under the Creative Commons Attribution License (CC BY 4.0), which means that the manuscript, images, and Supporting Information files will be freely available online, and any third party is permitted to access, download, copy, distribute, and use these materials in any way, even commercially, with proper attribution. For these reasons, we cannot publish previously copyrighted maps or satellite images created using proprietary data, such as Google software (Google Maps, Street View, and Earth). For more information, see our copyright guidelines: http://journals.plos.org/plosone/s/licenses-and-copyright.

Additional Editor Comments:

Dear Ms. Mary Wakeford,

I am delighted to inform you that the reviewers have completed their work on your manuscript entitled "Mesophotic benthic communities associated with a submerged paleoshoreline in Western Australia." We believe that your research has the potential to make a significant contribution to the field of marine ecology and biodiversity conservation.

After careful evaluation and consideration, we have decided to accept your manuscript for publication in our journal, subject to minor revisions. The reviewers have provided constructive feedback to enhance the clarity and quality of your work. We kindly request that you address the comments provided by the reviewers and resubmit your revised manuscript as soon as possible. We will expedite the review process and make a final decision as soon as possible upon receiving your revised manuscript.

We appreciate your contribution to our journal and thank you for selecting us as a platform to share your research findings.

Sincerely,

Dr. Shamim Ahmad

Reviewers' comments:

Reviewer's Responses to Questions

**Comments to the Author**

1. Is the manuscript technically sound, and do the data support the conclusions?

Reviewer #1: Yes

Reviewer #2: Yes

2. Has the statistical analysis been performed appropriately and rigorously? 

Reviewer #1: Yes

Reviewer #2: Yes

3. Have the authors made all data underlying the findings in their manuscript fully available?

Reviewer #1: Yes

Reviewer #2: Yes

4. Is the manuscript presented in an intelligible fashion and written in standard English?

Reviewer #1: Yes

Reviewer #2: Yes

5. Review Comments to the Author

Reviewer #1: The manuscript (MS) is very well written and new idea is developed. The MS records the Australian coastline 125 which is basically 125 m below present day sea level which is accepted to be formed during the LGM. The various aims to present map seafloor bathymetry of the KEF and its surroundings in high resolution using multibeam sonar; determining sediment grainsize, texture and broad composition using benthic grabs; characterizing the spatial distribution of benthic communities using towed video, and predicting the fine-scale spatial distribution of key habitats across study areas were undertaken. The study provides vital baseline for Australian coastline for effective management of biodiversity in Australia’s marine estate. On the northwest shelf in particular, increasing pressures from industry (including fishing, oil and gas exploration and development, shipping, and tourism) provide an impetus for evidence-based decision-making is well produced.

Reviewer #2: The present manuscript is very well written and properly framed. The authors have tried to put an excellent data for creating the baseline information regarding Australian coast during the LGM period. The use of latest modern techniques for figuring the suitability of the region at this depth is also a new approach. The findings helps to demarcate the various benthic forms in general however, the details of the forms based on sediment sampling is needed and this has been addressed by the authors effectively. The statistical approach considered in the study is also very efficiently used for giving direction to the study. Hence, the study is an outcome of vigorous work and patiently handled.

6. PLOS authors have the option to publish the peer review history of their article (what does this mean?). If published, this will include your full peer review and any attached files.

Reviewer #1: No

Reviewer #2: No

---

## [Author Response · Author response to Decision Letter 0]

16 Jul 2023

Reviewer 1: I have incorporated and addressed your suggestions into my revision. They were very helpful. Thank you.

Reviewer 2: I have incorporated and addressed your suggestions into my revision. Thank you for your help.

---

## [Editor Report · Decision Letter 1]

27 Jul 2023

Mesophotic benthic communities associated with a submerged palaeoshoreline in Western Australia<o:p></o:p>

PONE-D-23-07404R1

Dear Dr. M Wakeford,

We’re pleased to inform you that your manuscript has been judged scientifically suitable for publication and will be formally accepted for publication once it meets all outstanding technical requirements.

Kind regards,

Shamim Ahmad, PhD

Academic Editor

PLOS ONE

Additional Editor Comments (optional):

Dear M Wakeford,

I am writing to inform you that your paper titled " Mesophotic benthic communities associated with a submerged palaeoshoreline in Western Australia " has been accepted for publication in PLOS ONE journal. Congratulations on your valuable research findings and contributions to the field of marine ecology and conservation.

The abstract of your paper highlights the significance of Key Ecological Features (KEFs) in Australia's Commonwealth marine environment and emphasizes the importance of understanding the biodiversity and ecosystem functions associated with these features. Your study investigating the Ancient Coastline at ~125m Depth Contour (AC125) on the northwest shelf of Australia addresses the gaps in existing research and sheds light on the poorly understood aspects of this specific KEF.

Your findings regarding the distribution and composition of benthic communities, particularly the association of filter feeder organisms with pockets of consolidated hard substrate, provide valuable insights into the biodiversity of the region. Furthermore, your development of spatially continuous maps of predicted benthic habitat classes, based on depth and large-scale structural complexity of the seafloor, serves as an essential tool for characterizing biodiversity in each study area.

Your conclusion that the distinct coastline, if it exists, is now largely buried and does not provide a unique hard substrate habitat, is significant for understanding the dynamics of the AC125 KEF.

I have recommended your paper for publication/presentation in PLOS ONE journal.

With best compliments

Dr Shamim Ahmad
---

## [Editor Report · Acceptance letter]

3 Aug 2023

PONE-D-23-07404R1 

Mesophotic benthic communities associated with a submerged palaeoshoreline in Western Australia<o:p></o:p>

Dear Dr. Wakeford:

I'm pleased to inform you that your manuscript has been deemed suitable for publication in PLOS ONE. Congratulations! Your manuscript is now with our production department. 

Kind regards, 

on behalf of

Dr. Shamim Ahmad 

Academic Editor

PLOS ONE